# Impact of storage conditions on the stability and biological efficacy of *trans*-arachidin-1 and *trans*-arachidin-3

Ploy Khongrungjarat[1], Chonnikan Tothong[1], Chanyanut Pankaew[1], Suchada Phimsen[1], Nopawit Khamto[1], Nutthamon Kijchalao[1], Warissara Wongkham[1], Piyathida Wongkham[1], Wipaporn Chuaymaung[1], Adsadayu Thonnondang[1], Apinun Limmongkon[1,2]*

1 Department of Biochemistry, Faculty of Medical Science, Naresuan University, Phitsanulok, Thailand,
2 Centre of Excellence in Medical Biotechnology (CEMB), Faculty of Medical Science, Naresuan University, Phitsanulok, Thailand

* apinunl@nu.ac.th

## Abstract

Prenylated stilbenoids, particularly *trans*-arachidin-1 (Ara-1) and *trans*-arachidin-3 (Ara-3), have gained attention for their notable bioactivities and potential health-promoting properties. This study presents the first comprehensive investigation into the stability and biological efficacy of these compounds in both peanut hairy root culture crude extracts (PCE) and partially purified fractions derived from elicited peanut hairy root cultures. PCE stored at –20 °C and 4 °C maintained higher antioxidant capacity, total phenolic content compared to samples stored at room temperature. In cytotoxicity assays using SW480 colon cancer cells, the extract stored at –20 °C retained bioactivity with only minor changes in $IC_{50}$ values over three months, demonstrating superior stability under frozen conditions. Over a six-month period, partially purified fractions of Ara-1 and Ara-3 showed a time-dependent decline in compound content. However, Ara-3 maintained strong cytotoxicity against KKU-100 cholangiocarcinoma cells, while Ara-1 exhibited a significant loss in activity. These findings demonstrate that low-temperature storage, particularly at –20 °C, is crucial for preserving the chemical integrity and bioactivity of stilbenoid-rich extracts. The study underscores the importance of optimizing storage conditions to ensure consistent bioactivity, supporting the potential application of these compounds in the development of stable and effective pharmaceutical or nutraceutical products.

## Introduction

Stilbenes or stilbenoids are naturally occurring polyphenolic phytoalexins that plants—particularly grapes, peanuts, and berries—produce in response to environmental stress [1]. Among them, resveratrol is the most extensively studied for

---

**Data availability statement:** All relevant data are within the manuscript and its Supporting Information files.

**Funding:** This work was supported by the National Research Council of Thailand (NRCT) and Naresuan University 2025 [Contract number N42A680157]. The funders had no role in study design, data collection and analysis, decision to publish, or preparation of the manuscript.

**Competing interests:** The authors have declared that no competing interests exist.

health-promoting properties, including antioxidant, anti-inflammatory, antimicrobial, and anticancer activities [2]. Research has shown that structural diversity of stilbenes plays a crucial role in determining their absorption and bioavailability, which directly impact their physiological effectiveness [3]. Prenylated stilbenes, such as *trans*-arachidin-1 (Ara-1) and *trans*-arachidin-3 (Ara-3), have emerged as promising compounds of interest due to their potential health benefits. These compounds are being investigation for their antioxidant and anti-inflammatory effects, as well as their anticancer activity [4,5].

Hairy root culture established via Agrobacterium-mediated transformation offers an effective system for producing stilbene compounds. This biotechnological approach offers advantages in terms of improved yield and consistent quality of stilbenes, making it a promising platform for the scalable production of bioactive compounds. To further boost stilbene biosynthesis in peanut hairy root cultures, elicitor-based strategies can be applied. These involve stimulating the plant defense pathways through the addition of specific elicitors, leading to increased production of compounds such as Ara-1 and Ara-3. Our previous work demonstrated that a combination of chitosan (CHT), methyl jasmonate (MeJA), and cyclodextrin (CD) significantly enhanced the accumulation of both Ara-1 and Ara-3 in peanut hairy root cultures [6]. Additionally, Sharma et al. [7] reported that treatment with CD and $H_2O_2$ selectively enhance Ara-1 levels, while the combination of CD and MeJA specifically increased Ara-3 production.

In recent years, the focus on prenylated stilbenoids has increased due to their enhanced biological activities compared to non-prenylated counterparts, highlighting their potential as promising health-promoting compounds. To ensure their bioactivity is maintained over time, stability testing plays a crucial role in evaluating plant secondary metabolites. This involves examining the impact of environmental factors such as temperature, light, and pH, all of which can influence the structural integrity and functional efficacy of these compounds. Therefore, understanding the stability of stilbenes under various conditions is essential for their effective application in health-related products. As these metabolites are susceptible to degradation under unfavorable conditions, establishing optimal storage and handling strategies is crucial to preserve their stability and maximize their effectiveness. HPLC analysis of crude *Bacopa monnieri* extracts showed a significant reduction in saponin levels, including bacopaside I and bacoside A, under all storage conditions. Notably, samples maintained under long-term conditions at 30°C and 65% relative humidity (RH) preserved saponins more effectively than those stored under accelerated (40°C, 75% RH) or real-time room temperature conditions [8]. Similarly, a stability and antioxidant assessment of allicin-containing aqueous extracts from *Allium ursinum* revealed a decline in antioxidant activity within 24 hours, with a degradation half-life of 15 hours and 45 minutes at ambient temperature [9]. Stilbene stability studies in *Vitis vinifera* L. further revealed that *cis*-ε-viniferin is thermally unstable and sensitive to light-induced conversion [10]. In addition, *trans*-stilbenes such as astringin, isorhapontin, piceatannol, and isorhapontigenin from Norway spruce bark demonstrated high photosensitivity, undergoing rapid *trans*-to-*cis* isomerization under UV exposure and more slowly under fluorescent light [11].

Despite the increasing interest in prenylated stilbenes, no prior studies have investigated the stability of Ara-1 and Ara-3 under storage conditions. This work represents the first comprehensive study to evaluate their stability and biological activity in both peanut hairy root culture crude extracts (PCE) and partially purified fractions. PCE derived from elicitor-treated peanut hairy root cultures were stored under varying temperatures for up to three months and assessed for antioxidant potential, quantity of Ara-1 and Ara-3, and their effects on the viability of SW480 colon cancer cells. In parallel, partially purified fractions enriched in either Ara-1 or Ara-3 was monitored for changes in compound levels and cytotoxic effects on KKU-100 cholangiocarcinoma cells over a six-month period. The insights gained from this study are critical for understanding the storage parameters of prenylated stilbenoids and underscore the importance of optimizing storage conditions for their application in health-related products.

## Materials and methods

### Hairy root culture and elicitor treatment

Hairy root cultures derived from the peanut cultivar Kalasin 2 (K2-K599), previously transformed with *Agrobacterium rhizogenes* were utilized in this study [12]. Elicitation experiments were conducted using liquid cultures of the hairy roots. Four grams of root tissue were cultured in 500 mL Erlenmeyer flasks containing 200 mL of half-strength Murashige and Skoog (MS) [13] liquid medium (2.2 g/L MS basal medium powder) supplemented with 1.5% sucrose. The elicitation treatment consisted of a combination of 200 mg/L chitosan (CHT), 100 µM methyl jasmonate (MeJA), and 6.87 mM cyclodextrin (CD), CHT + MeJA + CD; as described by Chayjarung et al. [6]. Cultures were maintained on a rotary shaker at 150 rpm, incubated at 25 °C in complete darkness, and harvested after 72 hours.

### Preparation of hairy root culture medium extract for stability assessment

After 72 hours of cultivation, the culture medium was harvested, and the hairy root tissues were separated from the liquid phase. The culture medium was subjected to liquid–liquid extraction using an equal volume (200 mL) of ethyl acetate, repeated three times, and the organic phase was separated by decantation to ensure efficient extraction. The combined organic phases were concentrated under reduced pressure at 40 °C using a rotary evaporator (Büchi Rotavapor, Switzerland) to obtain the peanut hairy root culture crude extracts (PCE) for subsequent stability assays.

For the stability assessment, a single batch (lot) of the PCE was prepared, thoroughly homogenized, and subsequently aliquoted into three independent portions, which were then stored at −20 °C, 4 °C, and room temperature (RT) for a period of three months prior to analysis. This approach was employed to specifically evaluate the effect of storage temperature on extract stability while minimizing variability arising from independent extraction procedures.

### HPLC analysis of stilbene compounds

Stilbene compounds were analyzed by high-performance liquid chromatography (HPLC) following the method described by Limmongkon et al. [14]. PCE were subjected to reversed-phase chromatography using a Luna C18(2) column (5 µm, 100 Å, 250 × 4.6 mm; Phenomenex, CA, USA). Separation was achieved through gradient elution with a mobile phase consisting of acetonitrile and 2% formic acid in water (30:70, v/v), at a constant flow rate of 1.0 mL/min. Detection was carried out using a UV detector set at 306 nm for *trans*-resveratrol (Res) and at 340 nm for *trans*-arachidin-1 (Ara-1) and *trans*-arachidin-3 (Ara-3). Quantification of each stilbene compound was performed using calibration curves derived from their respective standards. Chromatographic signals from the photodiode array detector (PDA) were recorded as voltage output (V), which is proportional to UV absorbance at the selected wavelength.

### Antioxidant activity assay

The 2,2'-azinobis-(3-ethylbenzothiazoline-6-sulfonic acid) (ABTS) assay was performed following the method described by Re et al. [15]. In brief, the ABTS radical cation (ABTS$^{•+}$) was generated by oxidizing ABTS with potassium persulfate.

The assay was initiated by mixing 2 µL of the sample with 198 µL of the ABTS$^{•+}$ solution, followed by incubation in the dark for 6 minutes. The reduction in absorbance, indicating the scavenging of ABTS$^{•+}$, was measured at 734 nm using a microplate reader (Varioskan LUX, Thermo Scientific, USA). Antioxidant capacity was quantified using a Trolox standard curve and expressed as Trolox Equivalent Antioxidant Capacity (TEAC), in µmol Trolox/g crude extract.

### Total phenolic content assay

Total phenolic content was determined using the Folin–Ciocalteu method, as described by Singleton et al. [16]. Briefly, 2 µL of sample was mixed with 10 µL of Folin–Ciocalteu reagent, followed by the addition of 10 µL of 20% (w/v) sodium carbonate and 178 µL of distilled water. The mixture was incubated in the dark for 30 minutes, and the absorbance was measured at 765 nm using a microplate reader (Varioskan LUX, Thermo Scientific, USA). Results were expressed as gallic acid equivalents (GAE), in mg gallic acid/g crude extract, based on a standard curve prepared with gallic acid.

### Partial purification of peanut hairy root culture extract

The PCE used for partial purification was obtained from 2 liters of culture medium elicited with a combination of CHT+MeJA+CD. Partial purification of the Ara-1 and Ara-3 fractions was performed using silica gel column chromatography, following the method described by Chayjarung et al. [17]. The column was eluted with a gradient solvent system of acetone: hexane at ratios of 10:90, 40:60, and 70:30. Each eluted fraction was collected and subjected to thin-layer chromatography (TLC) for preliminary screening.

Fractions enriched in Ara-1 and Ara-3 were identified by comparing their HPLC retention times with those of purified Ara-1 and Ara-3 reference standards. For the stability assessment, a single batch (lot) of the selected partially purified Ara-1- and Ara-3-enriched fractions was thoroughly homogenized, and aliquoted, then stored at −20 °C for a period of six months prior to subsequent analyses. This approach was employed to evaluate long-term storage stability while minimizing variability arising from independent purification batches.

### Structural identification and characterization of *trans*-arachidin-1 and *trans*-arachidin-3

The High-Resolution Mass Spectrometry (HRMS) was conducted using an Agilent 6540 UHD Q-TOF mass spectrometer (Agilent Technologies, Santa Clara, CA, USA) in negative ESI mode. Operating conditions were nitrogen drying gas 10 L min$^{-1}$ at 350 °C, nebulizer 30 psig, capillary 3,500 V, skimmer 65 V, octopole RFV 750 V, and fragmentor 250 V. Collision-induced dissociation was performed at 10, 20, and 40 V with ultra-high-purity nitrogen, and spectra were collected over an *m/z* range of 100–1,000. The nuclear magnetic resonance spectroscopy (NMR) spectrum was recorded on a Bruker DRX-400 NMR spectrometer using acetone-$d_6$ as the solvent. The $^1$H and $^{13}$C NMR spectra were recorded at 400 MHz and 100 MHz, respectively. The spectra were calibrated using the signals of the residual undeuterated solvent at $δ$ 2.05 ppm for $^1$H NMR and 29.84 ppm for $^{13}$C NMR. Coupling constant (*J*) are reported in hertz (Hz). The splitting patterns are reported as follows: s (singlet), d (doublet), dd (doublet of doublets) and m (multiplet).

### Cell viability assay

SW480 colon cancer cells and KKU-100 cholangiocarcinoma cells were obtained from the American Type Culture Collection (ATCC) and the Japanese Collection of Research Bioresources (JCRB) Cell Bank (Osaka, Japan), respectively. Cells were cultured in Dulbecco's Modified Eagle's Medium supplemented with 10% fetal bovine serum (FBS; Gibco, New York, NY, USA). Streptomycin and penicillin (each 100 µg/mL) were freshly added, and the cells were incubated at 37°C in a humidified atmosphere containing 5% $CO_2$, and 95% air. All chemicals and solvents used were of analytical grade and obtained from Sigma-Aldrich (St. Louis, USA).

To evaluate the cytotoxicity effects of the extracts on the cancer cell lines, a cell viability assay was performed using 3-(4,5-dimethylthiazol-2-yl)-2,5-diphenyltetrazolium bromide assay (MTT) method. Cells were seeded at density $8 \times 10^3$ cells/well in 96-well plates and cultured for 24 hours. All treatment concentrations were expressed in µg/mL to maintain consistency across experiments conducted with different cell lines and to facilitate direct comparison of cytotoxic responses. The SW480 cells were continuously treated with the PCE (stock concentration 400 mg/mL) at five concentrations ranging from 200 to 3200 µg/mL for a period of 3 months, with a vehicle containing 0.8% DMSO used as the negative control. For KKU-100 cells, treatments were conducted for 6 months using PCE, Ara-1, and Ara-3 (stock solutions at 100 mg/mL) at five concentrations ranging from 100 to 500 µg/mL, with a vehicle containing 0.5% DMSO serving as the negative control. After 48 hours of treatment, cells were incubated with MTT solution (0.5 mg/mL) for 4 hours at 37°C. The MTT solution was then carefully removed, and the resulting formazan crystals were solubilized with DMSO. Absorbance was measured using a microplate reader (Varioskan LUX, Thermo Scientific, USA) at 540 nm. Cell viability was calculated as a percentage relative to the control group.

SW480 colon cancer cells were used to evaluate short-term changes in cytotoxic activity of the crude extract under different storage temperatures, whereas KKU-100 cholangiocarcinoma cells, which exhibit higher sensitivity to arachidin compounds, were selected to assess the long-term stability and bioactivity of partially purified fractions.

## Statistical analysis

All data were statistically analyzed using one-way analysis of variance (ANOVA) followed by the least significant difference (LSD) post hoc test, implemented in IBM SPSS Statistics version 23. Results are reported as the mean ± standard deviation (SD) from three independent biological replicates, with statistical significance set at $p < 0.05, 0.01, 0.001$.

## Results

### Elicitation of peanut hairy root culture

Following the 72-hour elicitation with a combination of chitosan (CHT), methyl jasmonate (MeJA), and cyclodextrin (CD); CHT + MeJA + CD, the culture medium from peanut hairy root cultures was extracted with ethyl acetate to obtain the peanut hairy root culture crude extracts (PCE). As shown in Fig 1, the culture medium exhibited a yellowish coloration after the 72-hour treatment compared to the initial state at 0 hour.

### Stability assay of peanut hairy root culture extract compared with storage temperature

**Antioxidant activity and total phenolic content.** To assess the stability of the PCE under various storage conditions, samples were stored at –20 °C, 4 °C, and room temperature-RT (30 °C, used as the reference) for a period of three months. Bioactivity assays were conducted monthly to assess the antioxidant stability of the extracts as demonstrated in Fig 2, Table 1, Supplementary Figure S1 and Table S1 in S1 File.

As illustrated in Fig 2a and Table 1, the PCE stored at RT showed a marked reduction in ABTS antioxidant activity beginning in the second month of storage. In contrast, samples maintained at 4 °C and –20 °C retained their antioxidant capacity without statistically significant differences compared to the initial measurement. After three months of storage, the antioxidant activities of the PCE stored at RT, 4 °C, and –20 °C were 420.18 ± 76.97, 485.42 ± 98.85, and 524.94 ± 59.70 µmol Trolox/g crude extract, respectively. These results suggest that refrigeration or freezing conditions (4 °C or –20 °C) are more effective in preserving the antioxidant stability of the PCE over time.

Total phenolic content under different storage conditions is shown in Fig 2b and Table 1. The extract stored at RT exhibited a notable decline in phenolic content beginning in the second month, consistent with the trend observed in antioxidant activity. Similarly, a significant reduction was observed in the sample stored at 4 °C and –20 °C by the second month.

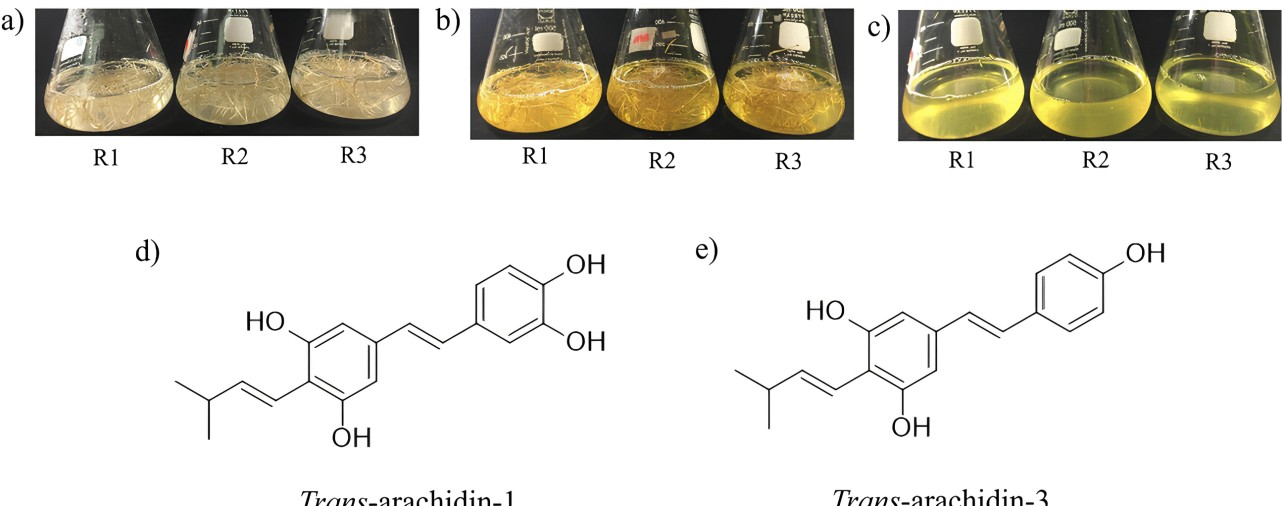

**Fig 1. Peanut hairy root cultures of the K2-K599 line treated with a combination of chitosan (CHT), methyl jasmonate (MeJA), and cyclodextrin (CD), shown in three biological replicates (R1, R2, R3).** (a) Hairy root cultures prior to elicitation (0 hour), (b) hairy root cultures after 72 hours of elicitation, (c) culture medium collected after 72 hours of elicitation. Chemical structures of the major stilbenoids: (d) *trans*-arachidin-1 (Ara-1) and (e) *trans*-arachidin-3 (Ara-3).

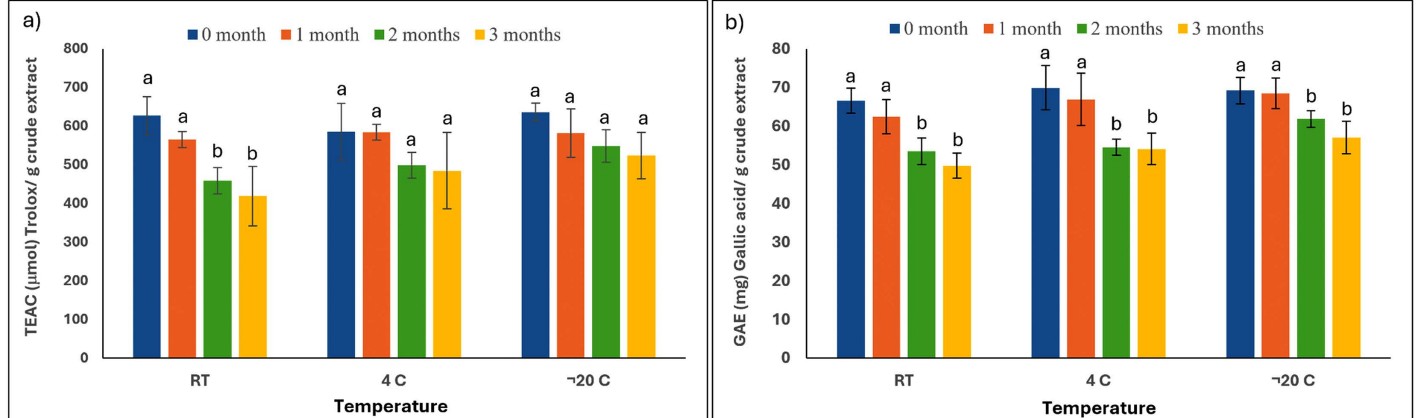

**Fig 2. Stability of (a) ABTS antioxidant activity and (b) total phenolic content in peanut hairy root culture crude extracts (PCE) stored at room temperature (RT), 4 °C, and –20 °C over a three-month period.** Samples were prepared from a PCE batch, aliquoted, and stored at the 3 months period prior to analysis. Bars represent mean ± SD (n = 3). Different letters within the same temperature condition indicate statistically significant differences ($p < 0.05$) among storage periods.

At the end of the third month, the phenolic contents of the extracts stored at RT, 4 °C, and –20 °C were 49.82 ± 3.22, 54.16 ± 4.06, and 57.15 ± 4.16 mg gallic acid/g crude extract, respectively.

**The predominant stilbenoids content.** HPLC analysis was conducted to assess the stability of the stilbenoids— *trans*-resveratrol (Res), *trans*-arachidin-1 (Ara-1) and *trans*-arachidin-3 (Ara-3)—in PCE subjected to different storage conditions. Although Res was included as a reference standard, it was not detected in either the crude extract or the partially purified fractions, indicating that its concentration was below the detectable level under the applied analytical conditions. The HPLC chromatograms of the non-elicited culture medium, used as the negative control for comparison

**Table 1. Stability assessment of antioxidant activity, total phenolic content, and stilbene compounds (*trans*-arachidin-1 and *trans*-arachidin-3) in peanut hairy root crude extracts (PCE) stored at room temperature (RT), 4 °C, and −20 °C over a three-month period. Different letters within the same column indicate statistically significant differences (p<0.05) among storage times under the same temperature condition.**

| Storage temperature | Storage times (months) | Antioxidant activity | Total phenolic content | Stilbene content | |
|---|---|---|---|---|---|
| | | TEAC (µmol trolox/g crude extract) | GAE (mg Gallic acid/g crude extract) | *Trans*-arachidin-1 (mmol/g crude extract) | *Trans*-arachidin-3 (mmol/g crude extract) |
| RT | 0 | 627.70±49.38 a | 66.66±3.25 a | 2.21±0.24 a | 13.54±0.51 a |
| | 1 | 566.09±20.65 a | 62.54±4.43 a | 2.02±0.08 a | 10.18±0.79 b |
| | 2 | 459.37±34.39 b | 53.54±3.47 b | 1.59±0.11 b | 9.77±0.26 b |
| | 3 | 420.18±76.97 b | 49.82±3.22 b | 1.59±0.11 b | 9.26±0.68 b |
| 4 °C | 0 | 586.13±73.73 a | 70.00±5.69 a | 2.26±0.19 a | 14.46±1.46 a |
| | 1 | 585.76±20.35 a | 66.99±6.77 a | 2.20±0.09 a | 11.91±0.55 b |
| | 2 | 499.50±33.70 a | 54.62±2.11 b | 1.93±0.15 b | 10.68±0.34 bc |
| | 3 | 485.42±98.85 a | 54.16±4.06 b | 1.70±0.12 b | 9.79±1.26 c |
| −20 ℃ | 0 | 637.48±23.47 a | 69.29±3.39 a | 2.32±0.20 a | 13.95±0.92 a |
| | 1 | 582.79±62.65 a | 68.56±3.96 a | 2.25±0.10 a | 12.93±1.21 ab |
| | 2 | 549.20±41.87 a | 61.95±2.12 b | 2.11±0.18 ab | 11.99±1.01 b |
| | 3 | 524.94±59.70 a | 57.15±4.16 b | 1.91±0.17 b | 10.15±0.47 c |

with elicited samples, are shown in Supplementary Figure S2 in S1 File. The HPLC chromatograms of the Ara-1 and Ara-3 standards are shown in Figs 3a and 3b, respectively. The freshly prepared PCE at 0 month (initial time point) (Fig 3c), together with samples stored for 3 months at RT (Fig 3d), 4 °C (Fig 3e), and −20 °C (Fig 3f), exhibited a gradual decline in the peak intensities of Ara-1 and Ara-3 under all storage conditions, indicating time-dependent degradation of these stilbenoid compounds. Notably, a new peak with a retention time of approximately 25 minutes appeared exclusively in the extract stored at RT (Fig 3d) but was absent in those stored at 4 °C and −20 °C.

The two predominant stilbenoids— Ara-1 and Ara-3—were quantified in the PCE over a three-month storage period. As shown in Fig 4a, Table 1, Supplementary Figure S3 and Table S2 in S1 File, the content of Ara-1 gradually declined across all storage conditions. A slight reduction was observed during the first month, followed by a more significant decrease in subsequent months. By the third month, samples stored at RT showed a substantial decline, with levels dropping to 1.59±0.11 mmol/g crude extract. Although a significant reduction in Ara-1 was evident in all samples by month three, storage at −20 °C provided the greatest stability, with only a minimal loss over the entire period. Samples stored at 4 °C exhibited a moderate decline, with statistically significant reductions observed from the second month onward in comparison to the measurements taken at month zero.

Similarly, the content of Ara-3 exhibited a time-dependent decline, as illustrated in Fig 4b, Table 1, Supplementary Figure S3 and Table S2 in S1 File. A significant reduction was evident as early as the first month in samples stored at RT, with concentrations decreasing from 13.54±0.51 to 9.26±0.68 mmol/g crude extract by the third month. Samples stored at 4 °C showed a significant decrease beginning in the first month. Although a decline was also observed under −20 °C storage, this condition preserved the highest residual levels, with Ara-3 content decreasing from 13.95±0.92 to 10.15±0.47 mmol/g crude extract over the same period. These results suggest that −20 °C is the most effective temperature for maintaining the stability of *trans*-arachidin compounds in PCE, whereas storage at RT accelerates their degradation.

## Viability assay of crude extract storage with SW480 colon cancer cells

The cytotoxic activity of peanut hairy root culture crude extract (PCE) against SW480 colon cancer cells following storage at RT, 4 °C, and −20 °C over a 3-month period was evaluated using MTT assays. Cytotoxic effects were initially assessed using concentration–response (% cell viability versus concentration) curves at each storage condition and time point, as

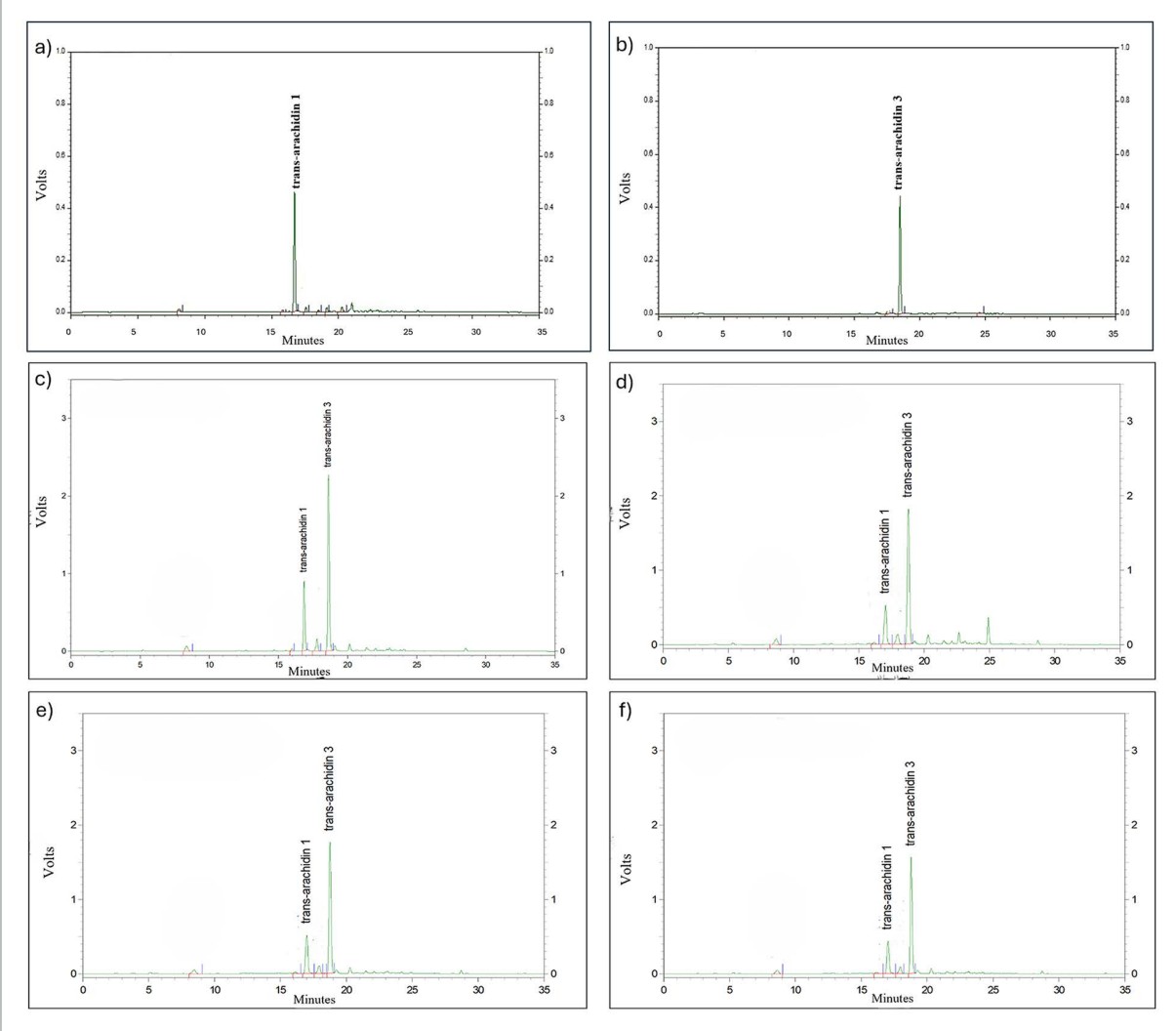

**Fig 3. HPLC chromatograms of (a) standard *trans*-arachidin-1 (Ara-1), (b) standard *trans*-arachidin-3 (Ara-3), and the peanut hairy root culture crude extracts (PCE) under different storage conditions: (c) freshly prepared extract at 0 month (initial time point), (d) extract stored for 3 months at room temperature (RT); (e) extract stored for 3 months at 4 °C; and (f) extract stored for 3 months at −20 °C.** Chromatographic signals were recorded as photodiode array detector voltage output (V), which is proportional to UV absorbance.

shown in Fig 5a–5c and Supplementary Figure S4 in S1 File. The corresponding $IC_{50}$ values were calculated from these curves and are summarized in Fig 5d and Supplementary Table S3 in S1 File.

At the initial time point (0 month), PCE induced a concentration-dependent reduction in SW480 cell viability compared with the vehicle control across all storage conditions. During storage at RT, a progressive loss of cytotoxic activity was observed, reflected by a rightward shift of the concentration–response curves and a reduction in statistical significance at equivalent concentrations over time (Fig 5a). Samples stored at 4 °C also exhibited a gradual decline in cytotoxic potency, although the magnitude of activity loss was less pronounced than that observed at RT (Fig 5b).

In contrast, PCE stored at −20 °C maintained relatively stable cytotoxic activity throughout the 3-month period, with concentration–response profiles comparable to those observed at the initial time point (Fig 5c). To quantitatively summarize these

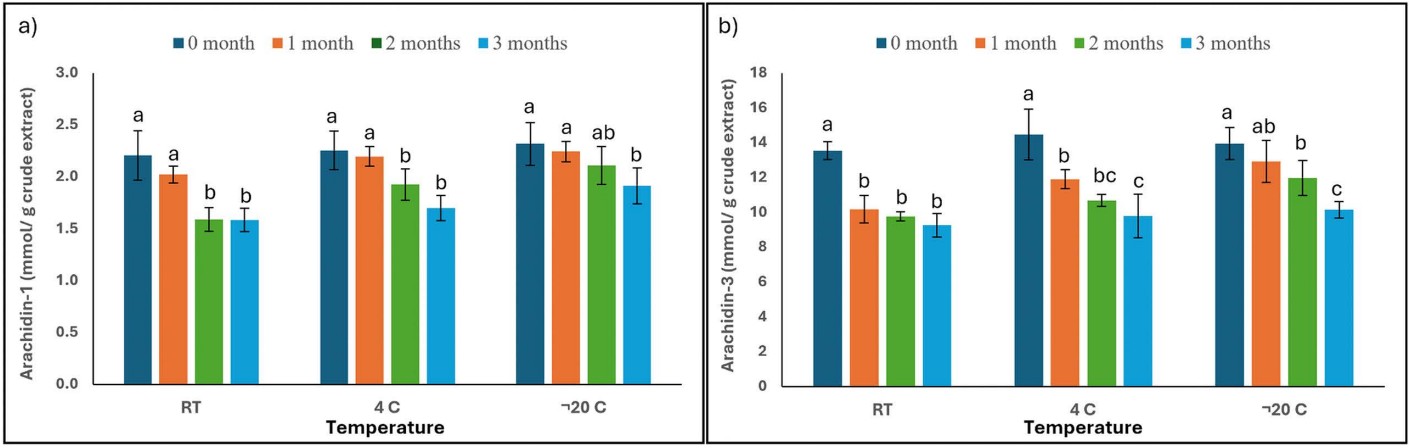

**Fig 4. Quantification and stability of (a) trans-arachidin-1 (Ara-1) and (b) trans-arachidin-3 (Ara-3) in peanut hairy root culture crude extracts (PCE) stored at room temperature (RT), 4 °C, and −20 °C over a three-month period. Stilbenoid contents were quantified by HPLC-UV analysis based on integrated peak areas and calculated using authentic Ara-1 and Ara-3 reference standards.** Data are presented as mean ± SD (n = 3). Different letters within the same storage temperature indicate statistically significant differences among storage time points (p < 0.05).

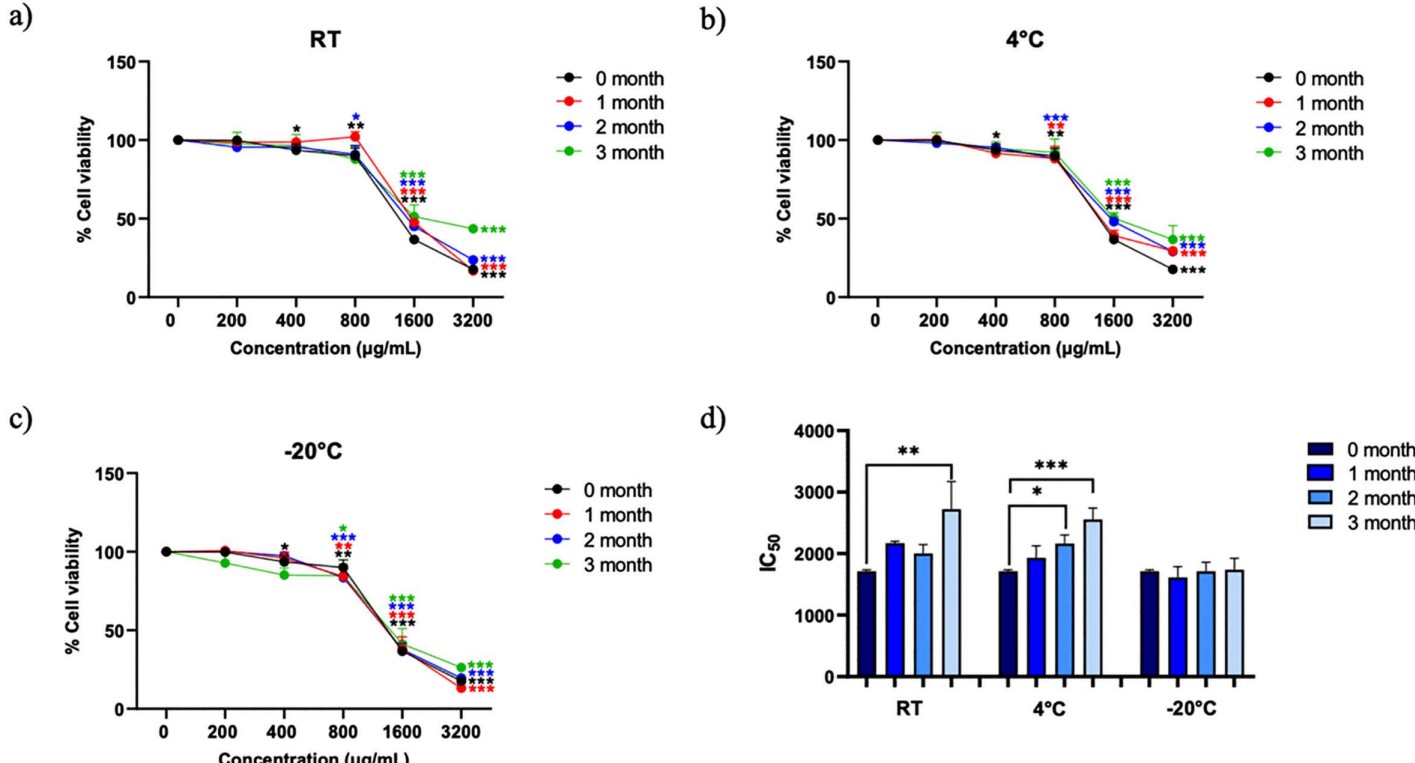

**Fig 5. Concentration–response effects of peanut hairy root culture crude extracts (PCE) on SW480 colon cancer cell viability following storage under different temperature conditions.** (a–c) show concentration–response curves expressed as percentage cell viability compared with the vehicle control after storage at room temperature (RT), 4 °C, and −20 °C, respectively, at 0, 1, 2, and 3 months. (d) summarizes the corresponding IC$_{50}$ values derived from these curves. Data represent mean ± SD from three independent experiments. In panels (a–c), *p < 0.05, **p < 0.01, and ***p < 0.001 indicate statistically significant differences compared with the vehicle control. In panel (d), *p < 0.05, **p < 0.01, and ***p < 0.001 indicates a statistically significant difference compared with the corresponding initial time point (0 month).

changes, $IC_{50}$ values were calculated from the concentration–response curves and are presented in Fig 5d. Statistical comparisons of $IC_{50}$ values were performed within each storage condition relative to the corresponding initial time point (0 month). Consistent with the concentration–response data, PCE stored at −20 °C showed no significant change in $IC_{50}$ values over time, whereas storage at RT and 4 °C resulted in increased $IC_{50}$ values, indicating a time-dependent reduction in cytotoxic activity.

Collectively, these results demonstrate that low-temperature storage, particularly −20 °C, effectively preserves the short-term cytotoxic activity of PCE in SW480 colon cancer cells. To further evaluate bioactivity stability in a more sensitive cancer model and to assess long-term storage effects, additional cytotoxicity assays were subsequently performed in KKU-100 cholangiocarcinoma cells.

## Stability assay of peanut hairy root culture partial purified fraction compared with crude extract

Based on the results, the PCE demonstrated good stability during the 3-month storage period at –20 °C. Therefore, a larger quantity of PCE was subjected to partial purification to isolate fractions enriched in Ara-1 and Ara-3 using silica gel column chromatography. High-resolution mass spectrometry (HRMS) and nuclear magnetic resonance (NMR) spectroscopy were employed to confirm the structural identity and characterize Ara-1 and Ara-3 in the partially purified fractions as demonstrated in Supplementary Fig S5-S10 in S1 FIle. The Ara-1 and Ara-3 were subsequently quantified in both the crude extract and the partially purified fractions. Furthermore, the stability of these compounds was evaluated over a 6-month storage period at –20 °C as demonstrated in Fig 6 and Supplementary Figure S11 in S1 FIle.

## Stability of *trans*-arachidin-1 and *trans*-arachidin-3 in partial purified fractions compared to crude extract (PCE)

The stability of Ara-1 and Ara-3 in the PCE was assessed over a 6-month storage period. As shown in Fig 6a, both compounds exhibited a time-dependent decline in content. The initial amount of Ara-1 was $0.72 \pm 0.07$ mmol/g crude extract, which gradually decreased over time. A significant reduction was observed beginning at month 3, with levels declining $0.45 \pm 0.10$ mmol/g crude extract by month 6. Similarly, Ara-3 content in PCE started at $1.52 \pm 0.21$ mmol/g crude extract and showed a moderate decline over the storage period. Significant reductions were noted at months 6, where the content decreased to $1.13 \pm 0.20$ mmol/g by the end of the 6-month period. These results indicate that both predominant stilbenoids in the PCE are susceptible to gradual degradation during storage, particularly after the third month.

Following partial purification, the Ara-1 content in the purified fraction was found to be higher than that in the PCE. The stability of Ara-1 in partial purified fraction was monitored independently for 6 months, as presented in Fig 6b. The initial amount was approximately $6.63 \pm 0.41$ mmol/g crude extract. A decrease was observed throughout the study, with a

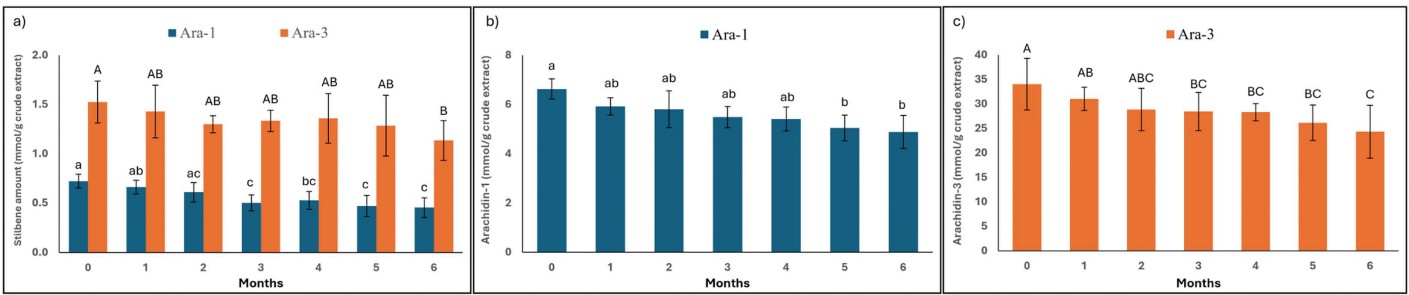

**Fig 6. Quantification and stability of predominant stilbenoids in (a) peanut hairy root culture crude extracts (PCE), (b) partially purified fraction enriched in *trans*-arachidin-1 (Ara-1), and (c) partially purified fraction enriched in trans-arachidin-3 (Ara-3) stored at −20 °C over a six-month period.** Stilbenoid contents were quantified by HPLC-UV analysis based on integrated peak areas and calculated using Ara-1 and Ara-3 reference standards. Samples were prepared from a purification batch, aliquoted, and stored at the indicated time points prior to analysis. Bars represent mean $\pm$ SD (n = 3). Different lowercase letters indicate statistically significant differences ($p < 0.05$) among storage periods for Ara-1, while different uppercase letters indicate statistically significant differences ($p < 0.05$) among storage periods for Ara-3.

significant reduction detected starting at month 5, and the level declined to 4.88 ± 0.67 mmol/g crude extract by month 6. The compound remained relatively stable compared to its form in PCE.

The stability profile of partial purified Ara-3 is shown in Fig 6c. The compound exhibited an initial amount of approximately 34.01 ± 5.25 mmol/g crude extract. A decreasing trend was observed, with levels declining steadily over the 6-month period. Significant differences were observed at months 3–6, reaching a final content of 24.29 ± 5.41 mmol/g crude extract by the end of the study. Although degradation occurred, the purified form exhibited a greater retention of stilbenoid content relative to the PCE, underscoring the stabilizing effect of the purification process.

## Stability of *in vitro* antioxidant activity and cytotoxic activity of KKU-100 cholangiocarcinoma cells

Due to the limited quantity of the partially purified fraction which was insufficient to support viability assays over the entire 6-month period, stability was evaluated at two representative time points (0 and 6 months). The *in-vitro* antioxidant activity of PCE, Ara-1, and Ara-3 was determined at 0 and 6 months of storage using the ABTS assay, and the results are presented in Fig 7 and summarized in Supplementary Table S4 in S1 File. A significant reduction in TEAC values was observed after 6 months for all samples ($p < 0.05$), with activity decreasing by approximately 1.2- to 1.3-fold compared with the initial levels. Among the samples, Ara-1 and Ara-3 exhibited higher antioxidant capacities than the crude extract, both at the beginning and after storage, indicating that these predominant stilbenoids contribute substantially to the antioxidant potential of the PCE and remain relatively stable even after long-term storage under low-temperature conditions.

In addition to the partially purified fractions, the crude extract (PCE) was also evaluated in KKU-100 cells at 0 and 6 months, allowing direct comparison between crude and purified samples under identical experimental conditions. The cytotoxic effects of PCE and the partially purified fractions of Ara-1 and Ara-3 in KKU-100 cholangiocarcinoma cell lines was assessed using concentration–response (% cell viability versus concentration) assays, with $IC_{50}$ values calculated as quantitative endpoints. The results were assessed based on percentage cell viability and $IC_{50}$ values as demonstrated in Fig 8, Supplementary Figure S12 in S1 File and summarized in Supplementary Table S5 in S1 File. At the initial time point (0 month), all treatments induced a concentration-dependent reduction in KKU-100 cholangiocarcinoma cell viability compared with the vehicle control (Fig 8a). Among the tested samples, Ara-3 exhibited the highest cytotoxicity potency, as indicated by the lowest $IC_{50}$ value, followed by Ara-1 and the PCE, indicating the superior potency of the purified compounds. After six months of storage, a reduction in cytotoxic activity was observed across all samples (Fig 8b). The $IC_{50}$ value of the PCE increased slightly from 295.03 ± 7.22 to 309.23 ± 57.32 μg/mL, with no statistically significant change in

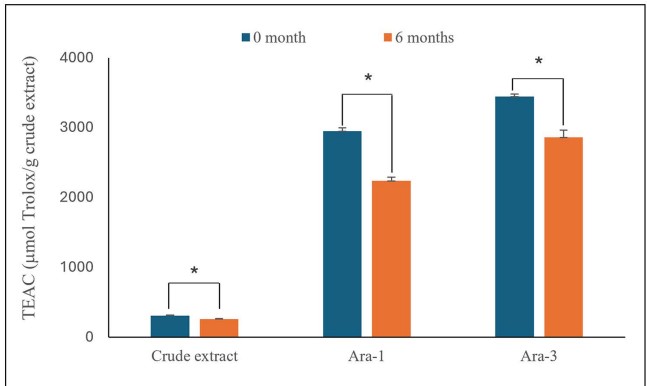

**Fig 7. Stability of ABTS antioxidant activity in peanut hairy root culture crude extract (PCE) and partially purified fractions of** *trans*-**arachidin-1 (Ara-1) and** *trans*-**arachidin-3 (Ara-3) after storage at −20 °C for six months.** Samples were prepared from a purification batch, aliquoted, and stored at the 6 months period prior to analysis. An asterisk (*) denotes a statistically significant difference ($p < 0.05$) relative to the initial measurement (0 month).

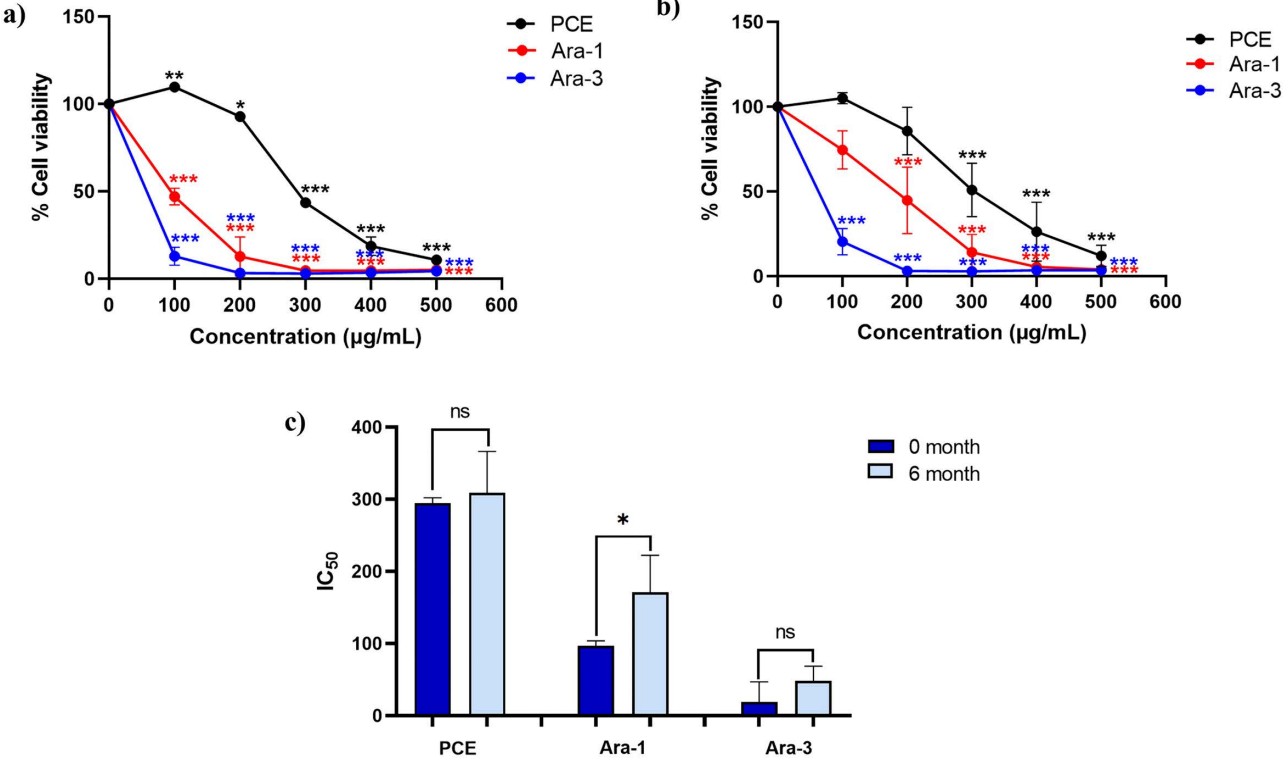

**Fig 8. Cytotoxic activity of peanut hairy root culture crude extract (PCE) and partially purified fractions of trans-arachidin-1 (Ara-1) and trans-arachidin-3 (Ara-3) against KKU-100 cholangiocarcinoma cells following storage at −20 °C.** (a) and (b) show concentration–response effects expressed as percentage cell viability at the initial time point (0 month) and after 6 months of storage, respectively. (c) presents a comparison of $IC_{50}$ values between 0 and 6 months for each sample. Data represent mean ± SD from three independent experiments. In panels (a) and **(b)**, *p < 0.05 and ***p < 0.001 indicate statistically significant differences compared with the vehicle control. In panel **(c)**, *p < 0.05 indicates a statistically significant difference compared with the corresponding initial time point (0 month), while ns indicates no significant difference.

activity. In contrast, the Ara-1 fraction showed a significant loss of cytotoxic potency, with $IC_{50}$ increasing from 96.66 ± 7.27 µg/mL to 171.30 ± 51.00 µg/mL. The Ara-3 fraction maintained a relatively stable bioactivity with an increase in $IC_{50}$ from 19.14 ± 27.77 µg/mL to 48.13 ± 20.66 µg/mL that remained statistically non-significant (Fig 8c).

Morphological changes and reduced cell density in KKU-100 cells following 48-hour treatments with the PCE and partially purified fractions at concentrations ranging from 100 to 500 µg/mL, after six months of storage as shown in Fig 9. Collectively, these results indicate that Ara-1 and Ara-3 are key contributors to the anticancer activity of the PCE. Among all tested samples, Ara-3 retained its cytotoxic efficacy most effectively over the six-month storage period, whereas Ara-1 showed a significant loss of activity. In contrast, the crude extract exhibited lower cytotoxic potency but greater stability over long-term storage.

## Discussion

Plant-derived stilbenoids have been increasingly recognized for their biological activity. In particular, prenylated stilbenoids such as *trans*-arachidin-1 (Ara-1) and *trans*-arachidin-3 (Ara-3) exhibit potent biological activities, including antioxidant [18], anti-inflammatory [5], and cytotoxic effects against various cancer cell lines [19]. Their unique structural modifications, such as the addition of prenyl groups, are believed to improve their membrane permeability and bioavailability, making them attractive for anticancer agent compared to non-prenylated analog of *trans*-reveratrol (Res). Studies comparing

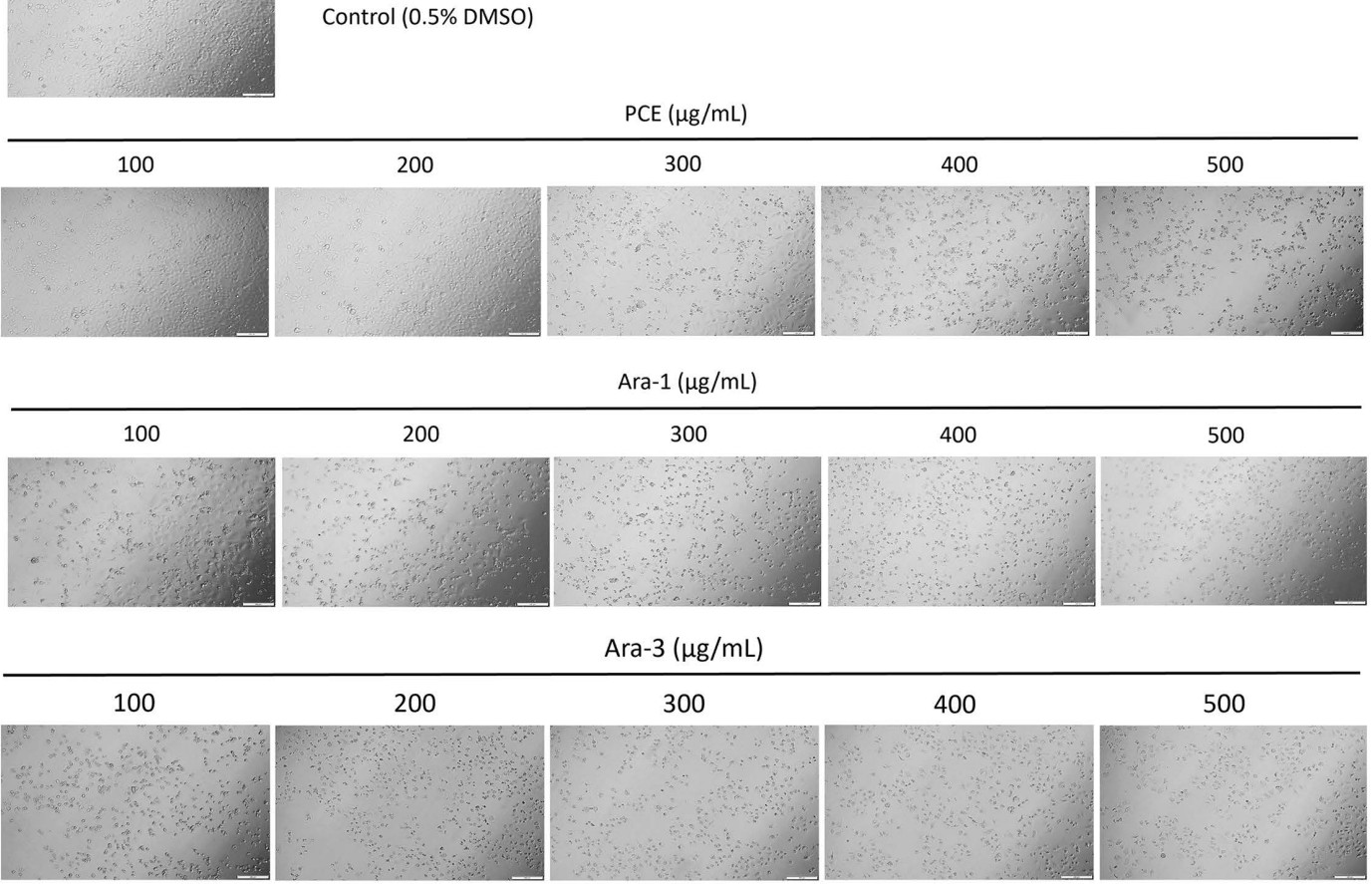

**Fig 9. Microscopic images of KKU-100 cholangiocarcinoma cells (10x magnification) after 48 hours of treatment with peanut hairy root culture crude extracts (PCE) and partially purified fractions of *trans*-arachidin-1 (Ara-1) and *trans*-arachidin-3 (Ara-3) at concentrations ranging from 100 to 500 μg/mL, after six months of storage.**

cytotoxic effects indicate that Ara-1 demonstrates superior anticancer activity relative to Ara-3 and Res in triple-negative breast cancer cells [20].

After 72 hours of elicitation, the hairy root culture medium exhibited a noticeable yellowish coloration compared to its initial state at 0 hour. This visible change is consistent with previous reports indicating that the secretion of stilbenoids into the medium often correlates with medium coloration [6]. In this study, the color change suggests the enhanced production and extracellular release of Ara-1 and Ara-3 in response to elicitor stimulation. The antioxidant activity observed in the peanut crude extracts (PCE) is therefore likely attributed to these two predominant stilbenoids—Ara-1 and Ara-3—, which were identified and quantified as the major bioactive constituents. Consequently, their stability during storage was further monitored by HPLC quantification to evaluate their contribution to the overall antioxidant efficacy. The relatively low variation observed in the quantified HPLC results across storage conditions can be attributed to the experimental design of this study. A single, well-homogenized batch of crude extract was prepared and subsequently aliquoted into independent three replicates prior to storage at −20 °C, 4 °C, and room temperature (RT) for three months. As all samples originated from the same extraction lot and were subjected to identical handling and preparation before storage, variability in HPLC

peak areas among replicates was minimized. This controlled approach enabled the isolation of storage temperature as the primary variable influencing extract stability, thereby allowing a more reliable assessment of temperature-dependent changes in stilbenoid content. Due to the significant bioactivity of stilbenoids, understanding their long-term stability under various storage conditions such as temperature and time is essential for maintaining the consistency and efficacy of bioactive extracts. The stability and functionality of phenolic compounds are highly dependent on a combination of environmental and structural factors, such as temperature [21], exposure to light [22], humidity [23], the nature of the surrounding matrix [24], and molecular structure [25]. Specifically, studies have shown that the stability of stilbene compounds such as Res declines at elevated temperatures [23]. The phenolic constituents in virgin olive oil exhibited improved stability and extended shelf life when stored at lower temperatures [26].

Our results demonstrated that PCE maintained at –20 °C retained higher levels of phenolic content and antioxidant activity as well as the Ara-1 and Ara-3 levels throughout a three-month period compared to those kept at RT. Additionally, the preservation of antioxidant activity at –20 °C paralleled the maintenance of cytotoxic activity against SW480 colon cancer cells, indicating a link between chemical stability and biological function. Although a gradual decrease in antioxidant activity was observed at −20 °C, the rate of decline was significantly lower than that at RT, indicating greater stability under low-temperature storage. The minor loss of activity may be attributed to slow oxidative degradation or light-induced structural alterations occurring during prolonged storage [27]. Our present study demonstrates that maintaining low storage temperatures is crucial for preserving stilbene stability, thereby supporting the retention of key bioactivities, including antioxidant and anticancer properties. The pronounced reduction in Ara-1 and Ara-3 content observed under RT storage is likely due to photo-oxidation and thermal degradation, processes commonly reported for stilbene compounds [28]. Exposure to light and oxygen can induce *trans–cis* isomerization of the stilbene double bond, followed by oxidative cleavage or polymerization, resulting in the formation of less active phenolic derivatives [29]. The presence of minor new peaks in the HPLC chromatograms supports this degradation pathway. The result suggest that low-temperature storage is more effective in preserving the chemical stability of stilbenoids in PCE over extended periods. This is in agreement with previous reports on stilbene photodegradation under UV and fluorescent light, which induces *trans*-to-*cis* isomerization and the formation of phenanthrene-type derivatives [11].

Given that low-temperature storage effectively preserves phenolic integrity and antioxidant capacity, we extended our investigation to assess the long-term stability of partially purified Ara-1 and Ara-3-enriched fractions over a six-month storage period. Compared to PCE, the purified fractions retained higher levels of the predominant stilbenoids—Ara-1 and Ara-3— suggesting that purification enhances chemical stability. This observation may be explained by the reduced influence of complex matrix components in purified samples, which can otherwise promote diverse degradation pathways. A similar phenomenon was observed in the degradation of pelargonidin-3-O-glucoside, where the compound degraded more rapidly in strawberry extract due to matrix complexity than in its purified form [30]. Based on preliminary observations indicating greater sensitivity of KKU-100 cholangiocarcinoma cells to arachidin compounds, the partially purified fractions were further evaluated for their cytotoxic activity against KKU-100 cells. Cytotoxicity assays of the partially purified fractions indicate that both Ara-1 and Ara-3 are major contributors to the anticancer activity observed in the PCE. Among the tested compounds, Ara-3 displayed the strongest cytotoxic effect, exhibiting the lowest $IC_{50}$ value, followed by Ara-1 and the PCE. Long-term stability testing demonstrated that Ara-3 retained its bioactivity more effectively than Ara-1, which showed a substantial decrease in cytotoxic potential after six months of storage. Although different cancer cell lines were employed in this stepwise experimental design, the consistent trends observed across models reinforce the robustness of the storage-dependent bioactivity findings.

Interestingly, a slight increase in cell viability was observed at the lower concentration (100 µg/mL) of PCE compared with the untreated control, as observed in Fig 5, suggesting a biphasic, or hormetic, effect. This biphasic, or hormetic, effect has been widely reported for polyphenolic compounds, where low concentrations stimulate cell proliferation or activate antioxidant defense mechanisms, while higher concentrations exert cytotoxic or pro-apoptotic effects. The mild

stimulatory response observed here may result from minor bioactive constituents within the crude extract that exert cyto-protective or antioxidant effects at sub-toxic levels. Similar dose-dependent responses have been reported for stilbenoids such as resveratrol and arachidin derivatives, which can activate the Nrf2/ARE-mediated antioxidant pathway, enhancing cellular defense and survival at low concentrations but inducing apoptosis at higher doses [4,5].

The stability and bioactivity of stilbenoids are strongly influenced by their structural features, including the degree and position of hydroxylation, methoxylation, and glycosylation, which affect compound degradation and their interaction with cellular targets [10,31]. Low-temperature storage (4 °C to −20 °C) is commonly employed in commercial formulations of natural antioxidant products to minimize oxidative degradation and prolong product stability, as previously reported for polyphenolic-rich extracts and resveratrol derivatives [32]. Therefore, the present findings provide commercially relevant insights into the preservation of Ara-1 and Ara-3 under storage conditions. To further mitigate degradation, strategies such as protection from light exposure, storage under an inert atmosphere, and the incorporation of stabilizing agents such as ascorbic acid or tocopherols are recommended to enhance the long-term stability of antioxidant activity [27]. Future studies should extend these findings by comparing the stability and antioxidant performance of Ara-1 and Ara-3 with established commercial antioxidant products under comparable storage and testing conditions to better assess their potential for practical applications.

## Conclusion

To our knowledge, this is the first report assessing the long-term stability of PCE from elicited peanut hairy root culture medium containing high levels of prenylated stilbenes. The results demonstrated that storage at low temperatures significantly preserved the chemical and biological properties of the extracts. After 3 months, samples maintained at –20 °C retained higher antioxidant activity, total phenolic content, and cytotoxic efficacy against SW480 colon cancer cells compared to those stored at room temperature. Furthermore, prolonged storage of partially purified Ara-3 for up to 6 months at –20 °C resulted in the highest cytotoxic activity against KKU-100 cholangiocarcinoma cells, followed by Ara-1 and the PCE. This enhanced biological activity was in line with the preserved chemical stability of the purified compounds. The data underscore the importance of low-temperature storage in maintaining both the structural integrity and bioactivity of stilbenoids during extended storage. From a future perspective, improving the bioavailability of stilbenes through advanced formulation techniques—such as encapsulation or sustained-release systems—may offer promising strategies to enhance their stability and functional efficacy for long-term storage and subsequent use in therapeutic and nutraceutical applications.

## Supporting information

**S1 File. Figure S1.** Line graphs showing the stability trends of (a) ABTS antioxidant activity and (b) total phenolic content in peanut hairy root culture crude extracts (PCE) stored at room temperature (RT), 4 °C, and −20 °C over a three-month period. **Figure S2.** HPLC chromatograms of (a) the culture medium without elicitation (negative control) and (b) peanut hairy root culture crude extracts (PCE). **Figure S3.** Line graphs showing the stability of (a) *trans*-arachidin-1 (Ara-1) and (b) *trans*-arachidin-3 (Ara-3) contents in peanut hairy root culture crude extracts (PCE) stored at room temperature (RT), 4 °C, and −20 °C over a three-month period. **Figure S4.** Concentration–response curves showing percentage cell viability of SW480 colon cancer cells treated with peanut hairy root culture crude extract (PCE) at increasing concentrations following storage at room temperature (RT), 4 °C, and −20 °C at different time points: (a) 0 month (initial); (b) 1 month; (c) 2 months; and (d) 3 months. **Figure S5.** High-resolution mass spectrum (ESI-MS/MS) spectrum and plausible fragmentations of *trans*-arachidin-1. **Figure S6.** High-resolution mass spectrum (ESI-MS) spectrum of *trans*-arachidin-3 **Figure S7.** $^1$H NMR spectrum (400 MHz, acetone-$d_6$) of *trans*-arachidin-1 **Figure S8.** $^{13}$C NMR spectrum (100 MHz, acetone-$d_6$) of *trans*-arachidin-1 **Figure S9.** $^1$H NMR spectrum (400 MHz, acetone-$d_6$) of *trans*-arachidin-3 **Figure S10.** $^1$H NMR spectra (400 MHz, acetone-d6) of *trans*-arachidin-1 (Ara-1) and *trans*-arachidin-3 (Ara-3). **Figure S11**. Line graphs showing the

stability of peanut hairy root culture extracts stored at −20 °C over a six-month period: (a) crude extract (PCE), (b) partially purified fraction enriched in *trans*-arachidin-1 (Ara-1), and (c) partially purified fraction enriched in *trans*-arachidin-3 (Ara-3). **Figure S12.** Cytotoxic activity of PCE derived from peanut hairy root culture media and partially purified fractions of *trans*-arachidin-1 (Ara-1) and *trans*-arachidin-3 (Ara-3) against KKU-100 cholangiocarcinoma cells following storage at −20 °C. (a–c) show concentration–response effects expressed as percentage cell viability at the initial time point (0 month), while (d–f) show the corresponding effects after 6 months of storage: (a, d) PCE; (b, e) partially purified fractions of Ara-1; and (c, f) partially purified fractions of Ara-3. **Table S1.** Replicate data ($N_1$–$N_3$) for ABTS antioxidant activity and total phenolic content of peanut hairy root culture crude extracts (PCE) stored at room temperature (RT), 4 °C, and −20 °C over a three-month period. **Table S2.** Replicate data ($N_1$–$N_3$) for of *trans*-arachidin-1 (Ara-1) and *trans*-arachidin-3 (Ara-3) contents in peanut hairy root culture crude extracts (PCE) stored at room temperature (RT), 4 °C, and −20 °C over a three-month period. **Table S3.** Replicate data ($N_1$–$N_3$) for $IC_{50}$ values of peanut hairy root culture crude extracts (PCE) against SW480 colon cancer cells following storage at room temperature (RT), 4 °C, and −20 °C over a three-month period. **Table S4.** Replicate data ($N_1$–$N_3$) for ABTS antioxidant activity of peanut hairy root culture crude extracts (PCE), and partially purified fractions of *trans*-arachidin-1 (Ara-1) and *trans*-arachidin-3 (Ara-3) stored at −20 °C, evaluated at 0 and 6 months. **Table S5.** Replicate data ($N_1$–$N_3$) for $IC_{50}$ values of peanut hairy root culture crude extracts (PCE) and partially purified fractions of *trans*-arachidin-1 (Ara-1) and *trans*-arachidin-3 (Ara-3) against KKU-100 cholangiocarcinoma cells following storage at −20 °C, evaluated at 0 and 6 months.
(RAR)

## Author contributions

**Conceptualization:** Ploy Khongrungjarat, Apinun Limmongkon.

**Data curation:** Chonnikan Tothong.

**Investigation:** Ploy Khongrungjarat, Chonnikan Tothong, Chanyanut Pankaew, Suchada Phimsen, Nopawit Khamto, Adsadayu Thonnondang.

**Methodology:** Ploy Khongrungjarat, Chonnikan Tothong, Chanyanut Pankaew, Nutthamon Kijchalao, Warissara Wongkham, Piyathida Wongkham, Wipaporn Chuaymaung.

**Project administration:** Apinun Limmongkon.

**Supervision:** Suchada Phimsen, Apinun Limmongkon.

**Validation:** Ploy Khongrungjarat, Nutthamon Kijchalao, Warissara Wongkham, Piyathida Wongkham, Wipaporn Chuaymaung.

**Writing – original draft:** Suchada Phimsen, Apinun Limmongkon.

**Writing – review & editing:** Nopawit Khamto, Apinun Limmongkon.

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
