## [Decision Letter · Decision Letter 0]

16 Oct 2025

PLOS ONE

Dear Dr. Limmongkon,

Thank you for submitting your manuscript to PLOS ONE. After careful consideration, we feel that it has merit but does not fully meet PLOS ONE’s publication criteria as it currently stands. Therefore, we invite you to submit a revised version of the manuscript that addresses the points raised during the review process.

We look forward to receiving your revised manuscript.

Kind regards,

Shengqian Sun

Academic Editor

PLOS ONE

Journal Requirements:

“This work was supported by National Research Council of Thailand (NRCT) and Naresuan University 2025 [grant number R2568A046].”

Reviewers' comments:

Reviewer's Responses to Questions

**Comments to the Author**

1. Is the manuscript technically sound, and do the data support the conclusions?

Reviewer #1: Yes

Reviewer #2: Yes

Reviewer #3: Yes

Reviewer #4: Yes

2. Has the statistical analysis been performed appropriately and rigorously?

Reviewer #1: N/A

Reviewer #2: Yes

Reviewer #3: Yes

Reviewer #4: No

3. Have the authors made all data underlying the findings in their manuscript fully available?

Reviewer #1: Yes

Reviewer #2: Yes

Reviewer #3: No

Reviewer #4: Yes

4. Is the manuscript presented in an intelligible fashion and written in standard English?

Reviewer #1: Yes

Reviewer #2: Yes

Reviewer #3: Yes

Reviewer #4: No

Reviewer #1: Manuscript ID: PONE-D-25-42817_reviewer

Title: Impact of storage conditions on the stability and biological efficacy of trans-arachidin-1 and trans-arachidin-3

The paper focuses on the stability and biological efficacy of these compounds in both peanut hairy root culture crude extracts and partially purified fractions derived from elicited peanut hairy root cultures. The paper’s overall structure is coherent, with a logical flow. The content of this MS is abundant. However, some issues are suggested to be addressed before publication. See my comments below to improve this manuscript.

1. This work is limited to laboratory research and lacks performance comparison with existing emergency antioxidant products.

2. The quality of the pictures needs to be improved. The picture is blurry and indistinct.

3. Which substances have antioxidant activity?

4. The antioxidant activity is generally affected at high temperature, and what is the commercial value of the study at low temperature? Please provide more comprehensive evidence or references in introduction and/or discussion.

In summary, this is a well-designed and well-conducted study. The authors are encouraged to refine and improve the manuscript based on the above suggestions to showcase the value of the research better and lay a foundation for future studies.

Reviewer #2: Comments

The manuscript “Impact of storage conditions on the stability and biological efficacy of trans-arachidin-1 and trans-arachidin-3” is interesting and brings out the aspect of storage conditions affect the properties of stilbenes. There are only few things that can be addressed to make it clearer.

Materials and methods

1. The intext citations need to be properly aligned like…Pilaisangsuree, Somboon (12) …. And others

2. Line 96… half-strength might see inappropriate, indicate the concentration

3. Line 104 can be rephrased to uindicate the technique of separation…fitration or decantation

4. Line 107…. rotary evaporator (Büchi)… is not clear is it the type of evaporator?

5. Line 130….also the references, and also indicate the equipment for reading absorbance

6. Line 143 …how did you determine the Fractions containing a predominant amount of Ara-1 and Ara-3

7. Concentrations…200- 3200 could be better written as 0.2-3.2mg/mL

8. Why the different concentrations of the DMSO

9. …. GraphPad Prism software version 9.4.0… should be in the analysis section

10.

Results

1. …This color change suggests…is well suited for the discussion part

2. For the

3. The major stilbene content is not clear

4. …This additional peak may represent…could be place d in the discussion section

5. The result section indicates the viability assay were done for three months and is not captured in the methods section clearly

6. In HPLC section you have trans-resveratrol (Res) which was not captured in other parts alongside trans-arachidin-1 (Ara-1) and trans-arachidin-3 (Ara-3).

Discussion

The discussion is well captured especially the photodegradation

Reviewer #3: The article " Impact of storage conditions on the stability and biological efficacy of trans-arachidin-1 and trans-arachidin-3" reported robust and diverse experiments to support their claims. Please address the following technical issues in a revised version:

1. Figure 2 a,b is it possible to represent these three month data points in bar graphs with individual data points included. As of now it is difficult to understand the spread of data points and these line graphs are very tightly packed, and error bars overlapping. Something to highlight the difference would be great. Although I highly recommend for ABTS assay, representing data in a table consisting of activity, km, kcat and kcat/km for efficient and clear presentation.

2. Figure 3. My suggestion is similar for product formation. Is it possible to integrate the peaks and normalize with the amount loaded on the column? Also, I am requesting a positive control (standard) and negative control (media minus products) if possible. The chromatogram traces are not really meaningful for comparison. Also statistics for hplc assay is not clear.

3. Figure 4, Figure 6 what assay was performed for stability testing? How are these figures different, except purification? Can any test of purity be shown?

4. Figure 5. and Figure 7. A control/sham group seems to be missing (cell lines' data without treatment) for both cell type data? Please check to add any relevant data.

Reviewer #4: In this manuscript, the authors investigated the effects of three storage conditions on the biological stability and activity of Ara-1 and Ara-3 isolated from PCE. Stability and activity were assessed using HPLC analysis, cytotoxicity assays, and antioxidant assays. Overall, the study is of interest; however, several issues should be addressed:

1. The authors reported that PCE stored at room temperature (RT) exhibited a greater decline in antioxidant activity compared with samples stored at 4 °C and –20 °C. However, a gradual decrease in antioxidant activity was also observed even at –20 °C. The authors should discuss possible reasons for this loss and propose strategies to minimize it.

2. In Figure 4b, samples stored at RT showed a marked reduction in Ara-3 content. The authors should clarify what degradation products are formed and describe the potential chemical mechanism underlying this process. Relevant reaction pathways or schemes would strengthen this discussion.

3. In Section 3.3.1, the authors present the stability of purified Ara-1 and Ara-3 over time under different conditions. What changes were observed in their antioxidant activities during storage?

4. In Figure 7, PCE treatment significantly increased cell viability at 100 µg/mL. The authors should provide an explanation or hypothesis for this observation.

5. The statistical analyses across figures are insufficiently described. Many figures lack p-value information, and the criteria for statistical significance are unclear. For instance, in Figure 7a, all three groups show significant differences at several concentrations—were these comparisons made versus untreated controls or between treatment groups?

**Do you want your identity to be public for this peer review?**  For information about this choice, including consent withdrawal, please see our Privacy Policy

Reviewer #1: **Yes:** Chen Yongsheng

Reviewer #2: No

Reviewer #3: **Yes:** Joydeep Chakraborty

Reviewer #4: No

---

## [Author Response · Author response to Decision Letter 1]

17 Nov 2025

The revised manuscript addresses all comments and suggestions provided by the reviewers and the academic editor as wriiten in the attached file: Response to Reviewers.

---

## [Decision Letter · Decision Letter 1]

18 Dec 2025

Dear Dr. Limmongkon,

Thank you for submitting your manuscript to PLOS ONE. After careful consideration, we feel that it has merit but does not fully meet PLOS ONE’s publication criteria as it currently stands. Therefore, we invite you to submit a revised version of the manuscript that addresses the points raised during the review process.

We look forward to receiving your revised manuscript.

Kind regards,

Shengqian Sun

Academic Editor

PLOS One

Journal Requirements:

Reviewers' comments:

Reviewer's Responses to Questions

**Comments to the Author**

Reviewer #2: All comments have been addressed

Reviewer #3: (No Response)

2. Is the manuscript technically sound, and do the data support the conclusions?

Reviewer #2: Yes

Reviewer #3: Partly

3. Has the statistical analysis been performed appropriately and rigorously?

Reviewer #2: Yes

Reviewer #3: I Don't Know

4. Have the authors made all data underlying the findings in their manuscript fully available?

Reviewer #2: Yes

Reviewer #3: (No Response)

5. Is the manuscript presented in an intelligible fashion and written in standard English?

Reviewer #2: Yes

Reviewer #3: Yes

Reviewer #2: The Author(s) have addressed the comments satisfactorily and i think the manuscript is okay in its current form.

Reviewer #3: I thank the authors for their kind consideration of the reviewer's comments. The manuscript is definitely shaping up better than previous submission. But presentation is not at par with publication standards. I urge the authors once again, to consider the following suggestions. Strictly aimed to concise redundant information and consider improving the figures with more traditional approaches for better understanding.

1. HPLC Y Axis units for PDA detector should be mAU? Please check.

2. Line 210: Where is table 1 in main text? Is it SI Table 1?

3. Line 225-227: "The freshly prepared PCE at 0 month (initial time point) (Figure 3c), together with samples stored for 3 months at RT (Figure 3d), 4 °C (Figure 3e), and −20 °C (Figure 3f), exhibited a gradual decline in the peak intensities of Ara-1 and Ara-3 under all storage conditions, indicating time-dependent degradation of these stilbenoid compounds". I still hold my reservation with this claim. The peaks were not integrated to calculate intensities. Is table S2 data relevant to this peak intensity calculation? If not, kindly provide. A LC chromatogram merely confirms time of retention and since extraction from replicates using organic solvents would be different, I expect a large variation in HPLC input samples. Normalizing against per gram crude extract may not suffice.

4. Line 240: i am again confused how Figure 4 and Table 1/S1 are related?

5. Figure 5: Can you kindly present this figure in a different way like a typical IC50 curve? At the moment it is incomprehensible. Instead of merely putting the IC50 numbers, please use %viability on Y and conc of compound on X and make 3 plots for RT, 4 and -20. The reason I suggest this is again, because the line curves are too close to each other for significance, unless your data is extremely reproducible, these are difficult to comprehend.

6. Results Section 3.3 - peak shift information are not needed in main text. Kindly consider omitting. In fact, please consider revision of this entire section and making the figure more informative. I cannot comprehend how this result section fits in your manuscript flow. Where is control data for NMR and also negative control?

7. Figure 7: Exactly what tests are performed here? Kindly update all legends with as much information as possible. When a reviewer is evaluating this manuscript it is the figure legends that help in understanding the figure. Going back and forth to the methods section is really an unfeasible approach.

8. Section 3.4.2 and Section 3.2.3 - why two different cell lines were chosen? Do you have any viability data related to Fig 5 for KKU-100 line? For consistency, can the cell data be considered to be represented together? The Figure 9 viability assay is well done and my suggestion was the same for Figure 5. Kindly consider a better flow of data for better understanding for readership.

**Do you want your identity to be public for this peer review?** For information about this choice, including consent withdrawal, please see our Privacy Policy

Reviewer #2: No

Reviewer #3: **Yes:** Joydeep Chakraborty

---

## [Author Response · Author response to Decision Letter 2]

21 Dec 2025

The response to reviewer has been attached as the "Response to reviewers2" file.

---

## [Editor Report · Decision Letter 2]

30 Dec 2025

Impact of storage conditions on the stability and biological efficacy of trans-arachidin-1 and trans-arachidin-3

PONE-D-25-42817R2

Dear Dr. Limmongkon,

We’re pleased to inform you that your manuscript has been judged scientifically suitable for publication and will be formally accepted for publication once it meets all outstanding technical requirements.

Kind regards,

Shengqian Sun

Academic Editor

PLOS One
---

## [Editor Report · Acceptance letter]

PONE-D-25-42817R2

PLOS One

Dear Dr. Limmongkon,

I'm pleased to inform you that your manuscript has been deemed suitable for publication in PLOS One. Congratulations! Your manuscript is now being handed over to our production team.

Kind regards,

on behalf of

Dr. Shengqian Sun

Academic Editor

PLOS One